# Evaluating Obstructive Sleep Apnea Utilizing Arterial Tonometry in Individuals with Cystic Fibrosis

**DOI:** 10.3390/arm93030020

**Published:** 2025-06-17

**Authors:** Michelle Chiu, Bethany Bartley, Elizabeth Gootkind, Salma Batool-Anwar, Donald G. Keamy, Thomas Bernard Kinane, Lael M. Yonker, Kevin S. Gipson

**Affiliations:** 1Department of Pediatric Pulmonology, Mass General Brigham for Children, Boston, MA 02114, USA; 2Department of Sleep Medicine, Brigham and Women’s Hospital, Boston, MA 02115, USA; 3Division of Sleep Medicine, Harvard Medical School, Boston, MA 02115, USA; 4Department of Otolaryngology-Head & Neck Surgery, Mass Eye & Ear, Boston, MA 02114, USA

**Keywords:** cystic fibrosis, obstructive sleep apnea, home sleep apnea testing, peripheral arterial tonometry

## Abstract

**Highlights:**

**What are the main findings?**
Home sleep apnea testing using peripheral arterial tonometry (PAT) identified a higher-than-expected prevalence of obstructive sleep apnea (OSA) in adults with cystic fibrosis (CF).Traditional OSA screening tools (e.g., STOP-Bang, ESS, PSQI) did not correlate with OSA diagnosis in this cohort of adults with CF.

**What is the implication of the main finding?**
Individuals with CF, especially those with reduced lung function or higher Mallampati scores, may be underdiagnosed for OSA.PAT-based home sleep testing may be a feasible, low-burden alternative to polysomnography for sleep evaluation in CF.

**Abstract:**

Poor sleep quality and excessive daytime sleepiness are commonly reported by individuals with cystic fibrosis. The potential impact of comorbid sleep-disordered breathing (SDB), particularly obstructive sleep apnea (OSA), has not been extensively studied in the CF population. At present, there are no specific recommendations available to help clinicians identify patients with CF who are at increased risk of sleep disorders. Home sleep apnea testing using a validated peripheral arterial tonometry (PAT) device may offer an accurate diagnosis of OSA in a more convenient and low-cost method than in-lab polysomnography. In this single-center study of 19 adults with CF, we found an increased prevalence of OSA among individuals with CF compared to general population estimates. Although associations with an FEV < 70% predicted and a modified Mallampati score ≥ 3 were observed, these odds ratios did not reach statistical significance, likely reflecting limited power in this small pilot sample. There was no association found between the self-reported presence of nocturnal cough or snoring and OSA. We also found no association between OSA and abnormal scores on commonly used, validated sleep questionnaires, suggesting that CF-specific scales may be needed for effective screening in the CF clinic.

## 1. Introduction

Cystic fibrosis (CF) is a progressive disease affecting approximately 30,000 people in the United States [1]. Recent advances in therapies, including cystic fibrosis transmembrane conductance regulator (CFTR) modulator medications, have dramatically improved morbidity and mortality from CF, and the median expected age of survival in the CF population has increased significantly [2]. In this new era of improved life expectancy, one area that remains under-examined is co-morbid sleep disorders in adults with CF.

Significant sleep issues have long been reported in both children and adults with CF, including snoring, excessive daytime sleepiness, frequent awakenings, and nocturnal hypoxemia, many of which may be associated with or suggestive of a diagnosis of obstructive sleep apnea (OSA) [3,4,5,6]. Prior studies have suggested a markedly increased prevalence of co-morbid OSA in children with CF of 55–75%, compared to the estimated prevalence in the general pediatrics population of 2–5% [7,8]. Additional studies have also found that OSA is an independent risk factor for morbidity and all-cause mortality and has a negative impact on inflammatory and immune processes in CF [9,10].

There are several barriers to the investigation of sleep issues in the CF population. First, no guidelines exist to help CF providers identify those at increased risk of sleep disorders. Second, the burden of care on individuals with CF remains significant, and in-laboratory polysomnography is associated with high costs and long wait times, which results in a meaningful barrier for many patients. Lastly, despite the overall poor subjective quality of sleep reported in individuals with CF and the known ramifications of untreated SDB, those with advanced lung disease may be less likely to seek medical attention for sleep issues than for other concerns [3]. Given this context, we aimed to assess the feasibility and clinical utility of home sleep apnea testing using a validated, FDA-cleared peripheral arterial tonometry (PAT) device in adults with CF. While PAT-based HSAT has been validated in general and select cardiovascular populations, its feasibility and performance have not been specifically evaluated in adults with CF, who may have unique physiological characteristics affecting device accuracy.

In this study, we utilized validated sleep questionnaires, a virtual upper airway exam to assign a modified Mallampati score (MMS), and a portable home sleep apnea test (HSAT) device using PAT to evaluate for OSA in adults with CF.

## 2. Materials and Methods

This pilot clinical feasibility study was a prospective cohort study of 19 adults (age ≥ 18 years, FEV1 percent predicted [FEV1pp] > 55%) with CF recruited from the CF Center at Massachusetts General Hospital. Participants were recruited via consecutive sampling of eligible individuals attending routine CF clinic visits. Subjects were required to be clinically stable for at least 2 weeks before the study (i.e., no hospital admissions or pulmonary exacerbations). The cohort included 10 female participants (53%) with a mean age of 35 ± 10 years and a mean FEV1pp of 78 ± 21%. All but one subject was taking CFTR modulator therapy. Specific demographic and clinical characteristics of the study participants are detailed in Table 1.

All study participants underwent HSAT with a PAT-based device previously validated as a viable alternative to polysomnography for diagnosis of OSA in individuals without severe lung disease [11]. This HSAT device, the WatchPAT ONE (ZOLL^®^ Itamar^®^, Caesarea, Israel), measures the following: PAT signal, oxygen saturation, actigraphy, snoring volume in decibels, chest movement, and body position. The overnight HSAT data are processed by proprietary software utilizing algorithms that detect respiratory and other events during sleep, including PAT-based device determination of the apnea–hypopnea index (pAHI). Notably, PAT with oximetry and actigraphy is considered technically adequate for the diagnosis of OSA by the American Academy of Sleep Medicine manual [12,13,14]. The HSAT studies were independently reviewed by two board-certified sleep physicians to ensure technical quality and adequacy of the study. All 19 HSAT studies with the PAT-based device were technically successful, with no data excluded due to poor quality. One study had a relatively abbreviated total recording time (TRT) of 3 h, 52 min, and an estimated total sleep time of 3 h, 7 min. The average estimated total sleep time across all 19 studies was 362 min (approximately 6 h).

All study participants completed three commonly used sleep questionnaires: the Epworth Sleepiness Scale (ESS), Pittsburgh Sleep Quality Index (PSQI), and STOP-Bang questionnaire.

Participants submitted standardized photographs of their upper airway with tongue protrusion, following instructions adapted from clinical MMS protocols. A pamphlet containing specific instructions and example pictures was provided to subjects, outlining the required images: (A) open mouth using a flash, (B) side profile against a plain background, and (C) wide smile with jaw closed using a flash. Three independent, blinded raters (study physicians) assigned modified Mallampati scores (I–IV) and tonsillar grading to each subject based on the open-mouth photo. These raters were blinded to the results of the PAT HSAT data at the time of photo assessment. Interrater reliability was assessed using Fleiss’ Kappa. For each binary exposure (e.g., FEV_1_ < 70% predicted, MMS ≥ 3, male sex), we compared the odds of OSA (defined as a pAHI > 5/h) versus no OSA. The OR was computed as (a × d)/(b × c), and 95% confidence intervals were calculated using the log odds method.

## 3. Results

Nineteen individuals with CF participated in the study. Ten participants were female (53%), with an overall mean age among study subjects of 35 ± 10 years, and mean FEV1pp was 78 ± 21% (Table 1). All but one subject was actively taking a CFTR modulator therapy.

In this cohort, the overall prevalence of OSA (pAHI > 5/h) was 32%. The prevalence of OSA by sex in study subjects was 44% (4/9) in males and 20% (2/10) in females. The odds of having a diagnosis of OSA was higher for an FEV1pp < 70 (OR 3.3, 95% CI [0.43, 26.0]) and male sex (OR 3.2, 95% CI [0.42, 24.42]). However, these findings did not reach statistical significance (Figure 1).

Positive scores for sleep questionnaires (STOP-Bang ≥ 3, ESS > 10, PSQI ≥ 7) were not associated with a pAHI > 5/h in our study population. On the PSQI, 37% of patients reported difficulty sleeping related to cough, and 37% of patients reported loud snoring during sleep. Neither the presence of nocturnal cough nor the presence of self-reported snoring was associated with OSA.

Data regarding PAT-estimated sleep architecture (average sleep efficiency 83% ± 9.6, average REM percentage 23% ± 7.3), as well as mean heart rate and oxygen saturation, were within normal limits for all participants.

The average MMS for all subjects was 2.6 ± 1.1. We found that an MMS ≥ 3 was associated with a pAHI > 5/h, though this finding was not statistically significant (OR 4.5, 95% CI 0.57, 35.52). Interrater reliability for MMS scoring was poor (Fleiss’ κ = 0.177, *p* = 0.009), suggesting limited consistency across raters.

## 4. Discussion

As poor sleep quality has significant effects on morbidity and mortality, there is a need for increased recognition of risk factors for co-morbid SDB in the CF population. OSA is the most common form of SDB, with the estimated prevalence of OSA in North America (with OSA defined as AHI > 5 events/h) cited as 15–30% in males and 10–15% in females [15]. Our small pilot study revealed a greater prevalence of OSA in both males and females with CF compared to the general population, with the prevalence of OSA (pAHI > 5) found to be 44% in males and 20% in females with CF in our study population. This study also suggests an increased risk of OSA in individuals with CF and an FEV1pp < 70% predicted or an MMS ≥ 3. Given our small sample size, these observed associations did not reach statistical significance and should be interpreted as preliminary, hypothesis-generating findings that require validation in larger cohorts.

Previous studies in the CF population have suggested that the most common complaints leading to referral for polysomnography are snoring and excessive daytime sleepiness [7]. In a notable contrast, our study found that in this CF cohort, neither snoring nor excessive daytime sleepiness (ESS > 10) was associated with the presence of OSA. Additionally, scores reflective of poor sleep quality or increased likelihood of OSA on the widely used PSQI and STOP-Bang questionnaires were not associated with SDB. These findings suggest that providers may need to consider factors beyond traditional OSA screening tools when considering further SDB evaluation in individuals with CF. The low agreement among raters in MMS scoring highlights the limitations of applying this assessment remotely via patient-submitted photographs. This variability may have contributed to the lack of significant association between MMS and OSA in this study.

This pilot study begins to identify risk factors for OSA in the CF population. The limitations of this study include a small sample size (*n* = 19), which limited our analyses and the statistical significance of our findings. While home sleep apnea testing using peripheral arterial tonometry has been validated in the general population, its application and specific utility in the adult cystic fibrosis population, where unique clinical considerations and barriers to traditional sleep studies exist, represents an important area of novel investigation. Our findings uniquely highlight the prevalence of OSA in this specific cohort and challenge the applicability of commonly used screening tools in this context. Although the accuracy of HSAT devices, including PAT, remains an area of active discussion, and while HSAT devices including PAT devices are currently recommended only for uncomplicated adult patients without significant pulmonary disease, prior studies have demonstrated good agreement between in-laboratory polysomnogram AHI and pAHI across FEV1 ranges in people with chronic obstructive pulmonary disease [13,16,17]. The use of PAT for home sleep apnea testing has been validated in general and cardiovascular populations, demonstrating a strong correlation with polysomnography. To our knowledge, no prior study has specifically evaluated the feasibility of PAT-based testing in adults with CF. Our findings suggest that PAT testing may be a practical alternative in this population, though further studies are needed to confirm validity. Despite the exploratory nature of these findings, the ease and safety of PAT-based HSAT may allow CF clinics to better identify individuals at risk for OSA and prioritize referral for diagnostic evaluation. This approach could help bridge gaps in access to sleep diagnostics, particularly for patients reluctant or unable to attend in-lab studies. Given its low burden and home-based nature, PAT-based HSAT offers a feasible, immediate first step for evaluating sleep-disordered breathing in CF patients within routine clinical care, though larger-scale validation studies will be important.

## 5. Conclusions

SDB is likely underdiagnosed in CF. As life expectancy for individuals with CF has increased dramatically with recent advancements in therapy including highly effective CFTR modulators, it is critical to address factors that impact morbidity and mortality. Future studies with larger sample sizes are needed to better determine attributes associated with increased risk of SDB to better guide providers in referring at-risk patients for formal sleep evaluation. HSAT devices can significantly lower existing barriers to referral, and single-use PAT-based devices offer a low-cost, accessible method of evaluation with a low infection risk—ideal for people with CF. We believe these findings are important not only for informing future study design aimed at assessing risk factors for SDB in the CF population, but also to help reduce barriers to sleep testing in the CF population.

## Figures and Tables

**Figure 1 arm-93-00020-f001:**
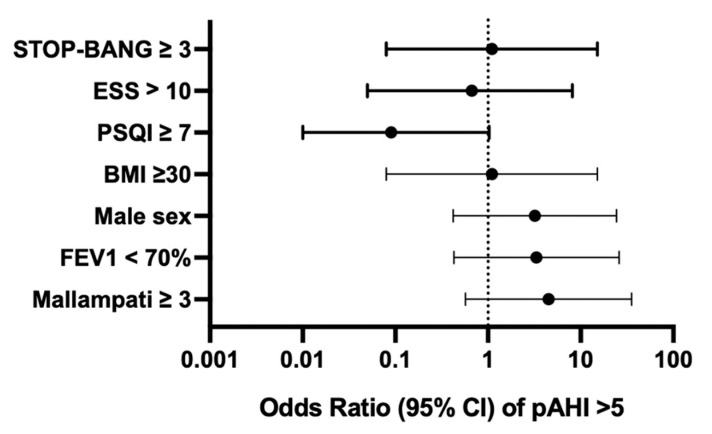
Forest plot of odds ratios of OSA (pAHI > 5/h) for various patient metrics. Odds ratios shown did not reach statistical significance; all 95% confidence intervals include 1.

**Table 1 arm-93-00020-t001:** Patient characteristics.

	All Subjects Mean (SD)	Male Subjects Mean (SD)	Female Subjects Mean (SD)
Age in years, mean (SD)	35 (10)	35 (10)	32 (20)
BMI in kg/m^2^	25.1 (4.5)	25.9 (5.2)	24.4 (3.9)
FEV1 percent predicted	78.1 (21.2)	76.3 (21.3)	76.6 (22)
ESS	6.5 (4.1)	6.1 (5.3)	6.9 (3)
STOP-BANG Score	1.5 (1.1)	2.1 (1)	0.8 (0.8)
PSQI	7.7 (3.6)	7.8 (4.1)	7.6 (3.3)

## Data Availability

Data collected in this study are available on request from the corresponding author due to institutional policy restricting public release of human subject data; all shared data will be de-identified.

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
