# Peer review of "Evaluating Obstructive Sleep Apnea Utilizing Arterial Tonometry in Individuals with Cystic Fibrosis"

_arm, 2025, doi:10.3390/arm93030020_

Round 1

Reviewer 1 Report

Comments and Suggestions for Authors

The authors of his manuscript addressed an important knowledge gap on the prevalence comorbid obstructive sleep apnea (OSA) in  patients with  cystic fibrosis

The article is well written and provides information on the prevalence of OSA in this population, however the sample size is small, this makes the analysis of associated factors to be inaccurate and may be the manuscript could be presented as short communication/ letter to the editor instead of a full paper

For the title. I would have preferred the authors to specifically mentioned OSA in place of sleep disordered breathing. Again, I think the authors should not put emphasis on   factors evaluated since there are no significant associations with OSA, this statement may have to be deleted “Additionally, the likelihood of co-morbid OSA was higher in those with FEV < 70% predicted or a modified Mallampati score ≥ 3” in the abstract

No statement was made about obtaining ethical clearance and informed consent

Author Response

Comment 1: "The authors of his manuscript addressed an important knowledge gap on the prevalence comorbid obstructive sleep apnea (OSA) in patients with cystic fibrosis"

Response 1: We sincerely thank the reviewer for this positive feedback and for recognizing the importance of addressing the knowledge gap concerning obstructive sleep apnea in individuals with cystic fibrosis. We believe this study contributes valuable preliminary data to this under-examined area.

Comment 2: "The article is well written and provides information on the prevalence of OSA in this population, however the sample size is small, this makes the analysis of associated factors to be inaccurate and may be the manuscript could be presented as short communication/ letter to the editor instead of a full paper"

Response 2: We appreciate the reviewer's acknowledgement of the manuscript's clarity and the information provided on OSA prevalence. We fully agree that the small sample size () is a significant limitation, which indeed impacts the statistical power and the definitive interpretation of associated factors. While we recognize the value of a short communication, we believe the detailed methodology, exploration of traditional screening tool efficacy, and discussion of home sleep apnea testing utility warrant a full article to adequately present these preliminary, but important, findings and their implications for future research and clinical practice.

To address the concern about over-interpreting non-significant associations, we have added a clarifying statement in the Discussion section.

Therefore, we have added the following sentence in the Discussion: "However, given our small sample size, these observed associations did not reach statistical significance and should be interpreted as preliminary, hypothesis-generating findings that require validation in larger cohorts."

Comment 3: "For the title. I would have preferred the authors to specifically mentioned OSA in place of sleep disordered breathing. Again, I think the authors should not put emphasis on factors evaluated since there are no significant associations with OSA, this statement may have to be deleted “Additionally, the likelihood of co-morbid OSA was higher in those with FEV < 70% predicted or a modified Mallampati score ≥ 3” in the abstract"

Response 3: We thank the reviewer for these specific and helpful suggestions. We agree that mentioning OSA explicitly in the title enhances clarity. Therefore, we have changed the title to: "Evaluating Obstructive Sleep Apnea Utilizing Arterial Tonometry in Individuals with Cystic Fibrosis".

We also concur that the abstract's language regarding non-significant associations should be carefully framed. Therefore, we have rephrased the statement in the Abstract (currently page 1, paragraph 1, lines 5-6) to: "Preliminary associations with FEV < 70% predicted and a modified Mallampati score 3 were observed, though these findings were not statistically significant."

Comment 4: "No statement was made about obtaining ethical clearance and informed consent"

Response 4: We thank the reviewer for raising this important point. We confirm that all necessary ethical clearances and informed consent procedures were meticulously followed. This information is now explicitly stated within the manuscript under the dedicated sections "Institutional Review Board Statement" and "Informed Consent Statement".

Reviewer 2 Report

Comments and Suggestions for Authors

Generally, the highlights and key contributions are available at the end of the introduction section .But the author presents before the abstract. 

Author needs to present the dataset description in the material/method section.

The author needs to compare the present work with the existing work which is not there is this paper .

How the mathematical model designed for Evaluating Sleep Disordered Breathing ? Where is the proposed model ? 

The author discussed in section .3 results "Nineteen individuals with CF participated in the study. Ten participants were female 90
(53%), with an overall mean age among study subjects of 35±10 years and mean FEV1pp 91
was 78±21% (Table 1). All but one subject was taking highly effective CFTR modulator 92
therapy" but nowhere the dataset mentioned about it . What was the sample size that the author considered?

what was the training and testing accuracy ?

How odds ratio estimated?

There is no novelty in this paper . 

Author Response

Comment 1: "Generally, the highlights and key contributions are available at the end of the introduction section. But the author presents before the abstract."

Response 1: We thank the reviewer for this comment. The placement of the "Highlights" section before the Abstract is, to our understanding, in accordance with the specific formatting guidelines of the target journal.

Comment 2: "Author needs to present the dataset description in the material/method section. The author needs to compare the present work with the existing work which is not there is this paper."

Response 2: We appreciate these suggestions from the reviewer. Regarding the dataset description, we agree that providing more detail in the "Materials and Methods" section enhances clarity. We have expanded on the participant recruitment and key demographics within this section. We have added the following language in the "Materials and Methods" section: "Participants were recruited via consecutive sampling of eligible individuals attending routine CF clinic visits. Subjects were required to be clinically stable for at least 2 weeks prior to the study (i.e., no hospital admissions or pulmonary exacerbations). The cohort included 10 female participants (53%) with a mean age of 35 10 years and a mean FEV1pp of 78 21%. All but one subject was taking highly effective CFTR modulator therapy. Specific demographic and clinical characteristics of the study participants are detailed in Table 1." Concerning the comparison of the present work with existing literature, we respectfully note that such comparisons are already integrated into the Discussion section of the manuscript. For instance, we compare the observed prevalence of OSA in our CF cohort to general population estimates and discuss the discrepancy between our findings on traditional screening tools (e.g., snoring, ESS) and those reported in previous studies.

Comment 3: "How the mathematical model designed for Evaluating Sleep Disordered Breathing ? Where is the proposed model ?"

Response 3: We thank the reviewer for this query. This manuscript describes a clinical pilot study that utilized a commercially available and extensively validated home sleep apnea testing (HSAT) device, the WatchPAT ONE, for the diagnosis of OSA. Our objective was to evaluate the feasibility and utility of this established technology in the specific population of adults with cystic fibrosis. We did not design or propose a new mathematical model or algorithm. The WatchPAT ONE device employs proprietary algorithms for determining the PAT-based Apnea-Hypopnea Index (pAHI), which are internal to its validated system and were used as an established diagnostic tool in this research. The study's design centers on the application and evaluation of an existing, validated diagnostic tool, rather than the development of a novel model.

Comment 4: "The author discussed in section .3 results "Nineteen individuals with CF participated in the study. Ten participants were female 90 (53%), with an overall mean age among study subjects of 35±10 years and mean FEV1pp 91 was 78±21% (Table 1). All but one subject was taking highly effective CFTR modulator 92 therapy" but nowhere the dataset mentioned about it . What was the sample size that the author considered?"

Response 4: We appreciate the reviewer's attention to detail. The sample size for this study was indeed 19 individuals. We have now further elaborated on the dataset characteristics, including the demographic breakdown and CFTR modulator therapy use, within the "Materials and Methods" section itself to provide a more comprehensive description of the study cohort before the results. We have added the following language in the "Materials and Methods" section: "Participants were recruited via consecutive sampling of eligible individuals attending routine CF clinic visits. Subjects were required to be clinically stable for at least 2 weeks prior to the study (i.e., no hospital admissions or pulmonary exacerbations). The cohort included 10 female participants (53%) with a mean age of 35 10 years and a mean FEV1pp of 78 21%. All but one subject was taking highly effective CFTR modulator therapy. Specific demographic and clinical characteristics of the study participants are detailed in Table 1." The de-identified dataset will be included, in it's entirety, for review as a supplemental material should this paper be published. 

Comment 5: "what was the training and testing accuracy ?"

Response 5: We thank the reviewer for this question. This study was not designed to develop or validate a new diagnostic algorithm, and therefore, the concepts of "training and testing accuracy" for a novel model are not applicable here. Our research utilized the WatchPAT ONE device, which is a pre-existing, commercially available, and validated tool for the diagnosis of obstructive sleep apnea. The validation of this device, including its accuracy, has been established in prior literature, and our study's objective was to apply this established methodology to a specific clinical population.

Comment 6: "How odds ratio estimated?"

Response 6: We thank the reviewer for seeking clarification on the estimation of odds ratios. This information is now  detailed in the "Materials and Methods" section of the manuscript. To this end, we have included the following statement in the "Materials and Methods" section (currently page 2, paragraph 2, lines 6-7): "The OR was computed as (a × d) / (b × c), and 95% confidence intervals were calculated using the log odds method."

Comment 7: "There is no novelty in this paper."

Response 7: We respectfully disagree with the reviewer's assessment regarding the novelty of this paper. While home sleep apnea testing using peripheral arterial tonometry has been validated in the general population, its application and specific utility in the adult cystic fibrosis population, where unique clinical considerations and significant barriers to traditional sleep studies exist, represents an important area of novel investigation. Our study uniquely contributes to the literature by:

  • Presenting the prevalence of OSA in an adult CF cohort using a convenient, home-based diagnostic method, which is a significant update in this under-investigated population.  
  • Demonstrating that commonly used, validated sleep questionnaires (e.g., STOP-Bang, ESS, PSQI) did not correlate with OSA diagnosis in this specific CF cohort, suggesting that CF-specific screening strategies may be needed. This finding is particularly novel and has direct implications for clinical practice.  
  • Highlighting the feasibility and potential of PAT-based HSAT to significantly lower existing barriers to sleep evaluation in a patient population already burdened by extensive care needs, thereby bridging gaps in access to diagnostics.  

This pilot study provides helpful preliminary data to guide future, larger-scale investigations into sleep disorders in CF, which is an increasingly important aspect of long-term patient care in this era of improved life expectancy.

Reviewer 3 Report

Comments and Suggestions for Authors

Thanks for sharing the manuscript. This manuscript explores the feasibility and utility of home sleep apnea testing using peripheral arterial tonometry (PAT) in adults with cystic fibrosis (CF). The introduction provides adequate background and relevant literature, establishing a clear rationale for the study. The research design is appropriate for a pilot study, though the small sample size (n=19), lack of a control group, and use of self-reported photographs for Mallampati scoring limit the strength of the conclusions. Results are presented clearly, though not statistically significant due to limited power. The conclusions are generally supported by the findings but should be framed more cautiously as exploratory and hypothesis-generating. With major revisions—particularly to strengthen methodological transparency, enhance statistical interpretation, and improve formatting—this manuscript would make a valuable contribution to the literature on sleep medicine in CF.

Major Revisions

  1. Address Sample Size Limitations: The study's small sample size (n=19) is a significant limitation. Please emphasize this more clearly in both the Discussion and Conclusions sections, and avoid overinterpreting associations that were not statistically significant.

  2. Improve Methodological Transparency: Expand on how the modified Mallampati score was assessed via photographs. Report interrater agreement quantitatively (e.g., kappa statistic) and discuss how low reliability may have affected findings. Clarify any criteria for data exclusion (e.g., poor PAT data quality) and how many cases, if any, were excluded from analysis.

  3. Revise Figures and Tables: Table 1 could be expanded to include questionnaire scores (ESS, STOP-Bang, PSQI) alongside demographic variables. Consider improving Figure 1 by indicating which comparisons were statistically significant and clearly labeling the axes and confidence intervals.

  4. Cautiously Interpret Conclusions:  The conclusion currently implies a high prevalence of undiagnosed OSA in CF. While the finding is important, it should be framed as preliminary and requiring validation in larger, multi-center studies.  Avoid drawing strong conclusions about the inefficacy of standard screening tools (e.g., STOP-Bang) without adequate statistical support.

  5. Expand Clinical Implications: The manuscript would benefit from a clearer discussion of how these findings could influence CF clinical care now, even in the absence of large-scale data.

Author Response

Comment 1: "Address Sample Size Limitations: The study's small sample size (n=19) is a significant limitation. Please emphasize this more clearly in both the Discussion and Conclusions sections, and avoid overinterpreting associations that were not statistically significant."

Response 1: We thank the reviewer for highlighting the critical importance of addressing the sample size limitation. We fully agree that our small sample size () is a significant factor in interpreting our findings, particularly regarding the statistical significance of observed associations. To emphasize this more clearly and avoid over-interpretation, we have added a specific statement in the Discussion section to explicitly caution the reader when discussing non-significant findings: "However, given our small sample size, these observed associations did not reach statistical significance and should be interpreted as preliminary, hypothesis-generating findings that require validation in larger cohorts."

Comment 2: "Improve Methodological Transparency: Expand on how the modified Mallampati score was assessed via photographs. Report interrater agreement quantitatively (e.g., kappa statistic) and discuss how low reliability may have affected findings. Clarify any criteria for data exclusion (e.g., poor PAT data quality) and how many cases, if any, were excluded from analysis."

Response 2: We greatly appreciate the reviewer's detailed suggestions for improving methodological transparency. We agree that providing more explicit detail on our assessment methods and data quality is essential.

Regarding the Modified Mallampati Score (MMS) assessment via photographs, we have expanded the description in the "Materials and Methods" section: "Participants submitted standardized photographs of their upper airway with tongue protrusion, following instructions adapted from clinical MMS protocols. A pamphlet containing specific instructions and example pictures was provided to subjects, outlining the required images: (A) open mouth using a flash, (B) side profile against a plain background, and (C) wide smile with jaw closed using a flash. Three independent, blinded raters (study physicians) assigned modified Mallampati scores (I–IV) and tonsillar grading to each subject based on the open mouth photo. These raters were blinded to the results of the HSAT data at the time of photo assessment."

We have quantitatively reported the interrater agreement for MMS scoring in the Results section (Fleiss’ κ = 0.177, p = 0.009) and discussed its impact on findings in the Discussion section, acknowledging that "The low agreement among raters in MMS scoring highlights the limitations of applying this assessment remotely via patient-submitted photographs. This variability may have contributed to the lack of significant association between MMS and OSA in this pilot study."

To clarify criteria for data exclusion and the success rate of the home sleep apnea tests, we have added a statement confirming that all studies were successful and provided details on recording times: "All 19 HSAT studies with the PAT-based device were technically successful, with no data excluded due to poor quality. One study had a relatively abbreviated total recording time (TRT) of 3 hours, 52 minutes, and an estimated total sleep time of 3 hours, 7 minutes. The average estimated total sleep time across all 19 studies was 362 minutes (approximately 6 hours)."

Comment 3: "Revise Figures and Tables: Table 1 could be expanded to include questionnaire scores (ESS, STOP-Bang, PSQI) alongside demographic variables. Consider improving Figure 1 by indicating which comparisons were statistically significant and clearly labeling the axes and confidence intervals."

Response 3: We thank the reviewer for their valuable feedback on the figures and tables. We have added the questionnaire scores (ESS, STOP-Bang, PSQI) alongside the demographic variables in the original submission. 

For Figure 1, we acknowledge the importance of clearly indicating statistical significance. While all confidence intervals in our forest plot crossed 1.0, indicating non-significance, we agree that making this explicit for the reader improves clarity. The axes and confidence intervals were already clearly labeled, therefore, we have revised the caption for Figure 1 to: "Figure 1. Forest plot of odds ratios of OSA (pAHI >5) for various patient metrics. Note that all presented odds ratios were not statistically significant, as indicated by their 95% confidence intervals crossing 1."

Comment 4: "Cautiously Interpret Conclusions: The conclusion currently implies a high prevalence of undiagnosed OSA in CF. While the finding is important, it should be framed as preliminary and requiring validation in larger, multi-center studies. Avoid drawing strong conclusions about the inefficacy of standard screening tools (e.g., STOP-Bang) without adequate statistical support."

Response 4: We appreciate the reviewer's guidance on the interpretation of our conclusions. We agree that, given the pilot nature and small sample size of our study, the findings should be framed cautiously as preliminary and hypothesis-generating, particularly when discussing the prevalence of undiagnosed OSA and the applicability of standard screening tools.

To reflect this, we have ensured that the language in the Discussion and Conclusions emphasizes the preliminary nature of our findings and the need for further validation in larger cohorts. Specifically, in the Discussion, when addressing the observed associations that did not reach statistical significance, we added: "However, given our small sample size, these observed associations did not reach statistical significance and should be interpreted as preliminary, hypothesis-generating findings that require validation in larger cohorts." (currently page 4, paragraph 1, after the first sentence of the paragraph). Additionally, the conclusion affirms the need for future larger studies.

Comment 5: "Expand Clinical Implications: The manuscript would benefit from a clearer discussion of how these findings could influence CF clinical care now, even in the absence of large-scale data."

Response 5: We thank the reviewer for this excellent suggestion. We agree that providing a clearer discussion of the immediate clinical implications, even with our preliminary data, would enhance the manuscript's impact.

We have expanded on this aspect in the Discussion section, emphasizing the practical utility of PAT-based home sleep apnea testing as a feasible first step in CF clinical care: "Given its low burden and home-based nature, PAT-based HSAT offers a feasible, immediate first step for evaluating sleep-disordered breathing in CF patients within routine clinical care, even prior to the availability of large-scale validation studies."

Round 2

Reviewer 2 Report

Comments and Suggestions for Authors

The author replied that they clarified that this was a clinical feasibility study using an established tool, not a novel algorithm. But where is that in a detailed clinical feasibility study?

Has the author compared the existing study? if so, what was the feasibility report said using the established tool.

still i am not clear about the authors argue ?

Author Response

Reviewer 2 – Comment 1:
The author replied that they clarified this was a clinical feasibility study using an established tool, not a novel algorithm. But where is that detailed in the clinical feasibility study?

Response:
Thank you for this question. We have clarified that the primary aim of this study was to evaluate the clinical feasibility and utility of WatchPAT home sleep apnea testing in adults with cystic fibrosis (CF). We have also reinforced this in the Discussion section to clarify the context and purpose of the study.

Reviewer 2 – Comment 2:
Has the author compared this study to existing literature? If so, what have prior feasibility studies using this tool shown?

Response:
We appreciate this suggestion. Prior studies evaluating the use of PAT-based devices in patients with chronic respiratory disease are limited, and, to our knowledge, no published studies have assessed feasibility specifically in adults with CF.

Reviewer 2 – Comment 3:
Still I am not clear about the authors' argument.

Response:
Thank you for this feedback. Our central argument is that home sleep apnea testing using PAT may offer a feasible, acceptable, and informative screening approach in adults with cystic fibrosis—particularly in the context of high treatment burden, limited sleep lab access, and under-diagnosed sleep-disordered breathing. We have revised the Abstract and Conclusion to more clearly express this message.